# Associations between the Home Physical Environment and Children’s Home-Based Physical Activity and Sitting

**DOI:** 10.3390/ijerph16214178

**Published:** 2019-10-29

**Authors:** Michael P. Sheldrick, Clover Maitland, Kelly A. Mackintosh, Michael Rosenberg, Lucy J. Griffiths, Richard Fry, Gareth Stratton

**Affiliations:** 1Research Centre in Applied Sports, Technology, Exercise and Medicine (A-STEM), Swansea University, Swansea SA1 8EN, UK; k.mackintosh@swansea.ac.uk (K.A.M.); g.stratton@swansea.ac.uk (G.S.); 2Centre for Urban Research, RMIT University, Melbourne, Victoria 3000, Australia; clover.maitland@rmit.edu.au; 3School of Human Sciences (Exercise and Sport Science), University of Western Australia, Perth 6009, Australia; michael.rosenberg@uwa.edu.au; 4Health Data Research UK, Wales and Northern Ireland, Swansea University Medical School, Swansea SA2 8PP, UK; lucy.griffiths@swansea.ac.uk (L.J.G.); r.j.fry@swansea.ac.uk (R.F.)

**Keywords:** house, moderate-to-vigorous physical activity, families, youth, objective, standing, screen-time

## Abstract

It is important to understand the correlates of children’s physical activity (PA) and sitting at home, where children spend significant time. The home social environment has an important influence; however, much less is known about the home physical environment. Therefore, the study aimed to assess relationships between the physical environment and children’s sitting and PA at home. In total, 235 child-parent dyads were included in the analyses. Children spent 67% of their time at home sitting. Linear regression analyses examined associations between physical home environmental factors obtained via an audit and children’s (55% girl, 10.2 ± 0.7) objective PA and sitting at home. Following adjustment for socio-demographics and social environmental factors, an open plan living area (OPLA), musical instrument accessibility and availability, and perceived house size were negatively and positively associated, whereas media equipment accessibility and availability was positively and negatively associated with sitting and standing, respectively. Additionally, an OPLA was positively associated with total and moderate-to-vigorous PA. Furthermore, sitting breaks were positively associated with objective garden size and negatively associated with digital TV. The physical home environment may have an important influence on children’s sitting, standing and PA at home; therefore, interventions that target this environment are needed.

## 1. Introduction

The importance of physical activity (PA) for children’s physiological and psychological health has been well documented [1], yet few meet current moderate-to-vigorous physical activity (MVPA) recommendations [2]. Children also spend most of their discretionary time in sedentary behaviours (7–8 h daily) [3], defined as ‘any waking activity, in a sitting, lying or reclining posture with an energy-expenditure below 1.5 metabolic equivalents (METs)’ [4]. Screen-time is the most prevalent sedentary behaviour, and has been associated with poor health outcomes [5]. However, the relationship between overall sedentary time and health in children is less clear [5]. Nonetheless, there is strong evidence for an adverse association between excessive levels and mortality in adults [6]. Recently, breaks from prolonged sitting have been beneficially associated with markers of body composition and metabolic health in adults [7]. Given the harmful consequences in adults and that children’s sedentary time appears to track into adulthood [8], high levels in children are a public health concern. Therefore, it is important to develop interventions to increase children’s PA and reduce their sedentary time.

Investigating the correlates of PA and sedentary time is essential for informing effective evidence-based interventions [9]. The social ecological model is often used to guide the understanding of children’s PA and sedentary time, recognising the important influence of the environment [10]. This model suggests that behavioural correlates are domain-specific, whereby behaviours are most likely influenced by the environment in which they occur [10,11]. Outside of school hours, children have been shown to spend significant time at home [12,13]. Indeed, there is also evidence indicating that a large proportion of children’s sedentary time and PA occurs at home [14,15,16]. Specifically, Tandon et al. [16] found that 48 and 42 % of children’s overall sedentary time and MVPA, respectively, was accumulated at home. The home environment, therefore, may be influential in affecting children’s PA and sedentary behaviours.

There is a plethora of research demonstrating the importance of the home social environment on children’s PA and sedentary time [17,18]. However, much less is known about physical environmental factors at home. Media equipment in the home and bedroom has consistently been positively associated with screen-time, but not overall sedentary time [17,18]. Additionally, there is some evidence that PA equipment is positively associated with PA [16,19] and inversely related to sedentary time [16,17]. Furthermore, whilst PA at home is most likely to occur outdoors [20], whether greater garden space facilitates PA remains unclear, with equivocal findings [21,22]. Even though there is an emerging evidence base, findings have been inconsistent, and research has been limited by the use of self-report instruments to measure the home and through assessing PA and sedentary time across the entire day [17]. Given behaviours are most likely influenced by the setting in which they occur [10], investigating sedentary time and PA at home will enable more precise identification of correlates. The use of objective measures such as audits and geographic information system (GIS) software to assess the home will also improve measurement accuracy. Additionally, greater media equipment accessibility has been shown to be associated with increased screen-time [19]; however, most studies have only assessed equipment availability.

When at home, children spend most of their time indoors [12,13]. This is of concern, because this is where children are most likely to be sedentary [20]. The indoor space may also be relevant for PA, with an ecological momentary assessment study showing that 30% of children’s aged 9–13 years leisure time PA occurred at home indoors [15]. Yet, few studies have explored influences on sedentary time and PA within the home indoor physical environment, outside of equipment [17]. A qualitative study identified several previously unexplored indoor physical environmental factors as potential influences on children’s sedentary time and PA at home, including multiple indoor living areas designated for screen-time, the presence of an open plan living area, the availability and layout of indoor space, as well as furniture within the home [23]. Additionally, new electronic media technologies such as online TV/movie streaming services may also be relevant, with just over 11 million people in the UK now being subscribed to one, as TV viewing habits shift online [24]. Moreover, playing musical instruments is an activity that commonly occurs at home [25], which can be done while sitting or standing [26]. Furthermore, houses with more than one floor may have a favourable effect on PA via increased stair climbing [27,28]. Exploration of the role of the factors cited by Maitland et al. [23], as well as musical instruments, movie/TV streaming services and the number of floors in influencing children’s sedentary time and PA is needed.

The aim of this study was to investigate the relationship between characteristics of the physical home environment and children’s home-based sitting, PA, standing and sitting breaks.

## 2. Materials and Methods

### 2.1. Study Design

The HomeSPACE study is a cross-sectional observational study investigating the influence of the home environment on children’s PA levels and sedentary time. Between November 2017 and July 2018, 235 children aged 9–12 years and their parents (*n* = 228) (response rate 26%) were recruited through primary schools from four of the largest conurbations in South Wales, Swansea (*n* = 174), Bridgend (*n* = 37), Cardiff (*n* = 16) and Newport (*n* = 8). A target sample size of 235 was set based on a reliable formula [29], while accounting for the possibility of missing data.

### 2.2. Recruitment

Primary schools (*n* = 23) were invited to participate. Eleven schools (response rate 48%) consented and 890 children aged 9–11 years were provided with information about the study. To be eligible, children had to be aged 9–12 years and without a physical disability. A chance to win a family pass for an outdoor adventure centre and the child’s sitting and PA results were offered as incentives. Informed consent and child assent were provided. The Swansea University ethics committee granted ethical approval for the study.

### 2.3. Home Physical Environment

HomeSPACE-II, an updated version of the HomeSPACE-I [30] and the Physical Activity and Media Inventory [31], was administered to the parents. The audit assessed physical home environmental factors hypothesised to influence children’s home-based PA and sedentary behaviours [23]. Parents were asked to walk around their house and garden and complete the items for each room/area. Briefly, the audit allowed the presence, amount and accessibility of 41 media (e.g., TV, computer, etc.), musical (e.g., drums, piano, etc.), PA (e.g., balls, trampoline, etc.) and seated furniture (e.g., sofa, desk etc.) items to be recorded for up to 22 room/areas (14 indoor and eight outdoor). Accessibility of each item was rated on a scale of A–D [31]. The response options were; A: put away and difficult to get to; B: put away and easy to get to; C: in plain view and difficult to get to; D: in plain view and easy to get to. There were questions relating to home features (house size, garden size, type of house, number of floors) and electronic media (smartphones, TV service, movie/TV streaming service). In addition, there were questions referring to the space to play inside the house, and in the back and front garden [30]. The audit data were reduced to several independent variables. Three dichotomous variables were generated to reflect the presence of: (1) an open plan living area; (2) a TV in the primary child’s bedroom; (3) a detached house. Yes and no responses were coded as 1 or 0, respectively. The number of living areas in the home with a TV was also calculated. In addition, summary scores that accounted for the accessibility and availability of PA equipment, seated furniture, overall media equipment, media equipment in the child’s bedroom and musical instruments were created by multiplying each item by its accessibility rating (A = 1, B = 2, C = 3, D = 4). A higher score indicates a greater overall “presence” of that type of item in the home. For descriptive purposes, we also calculated the total number of each type of item and the number of rooms/areas. Active video game systems (e.g., Wii Fit, Xbox Kinect, PlayStation move) were coded as PA equipment. Instruments were checked for missing data and for clarity, and followed up with families when needed.

### 2.4. Home Log Diary

Parents were given a diary to record when the child was at home each day for seven days, to allow for the calculation of home-based behaviours. Instructions were provided, where “Home” was defined as a single location, including the house, garden, driveway and verge of the home where the child spends most of their time (i.e., excluding homes of other parents). To minimise missing data, children completed the diary when parents were unable to and incomplete diaries were followed up with families.

### 2.5. Objectively Measured Home-Based Physical Activity and Postural Behaviours

Children wore the ActiGraph GT9X (ActiGraph, Pensacola, FL, USA) and the activPAL3 micro (PAL Technologies, Glasgow, UK), which measured total physical activity (TPA) and MVPA as well as postural behaviours (i.e., sitting, standing and sitting breaks), respectively, for seven consecutive days. A siting break was defined as a transition from sitting to standing/stepping [4]. Both were fitted at school, to ensure correct attachment and to provide instructions on how to reattach them. Participants were asked only to remove the monitors for swimming. Parents were also required to record sleep and wake times, device removals and any illness days.

The activPAL has demonstrated excellent validity in children [32], and was placed in a waterproof nitrile sleeve and secured on the midline of the upper right thigh using a hypoallergenic dressing (3M Tegerderm or Hypafix Transparent). Supplementary dressings, sleeves and instructions on correct reattachment were provided. ActivPAL data were downloaded using the manufacturer software (V8.10.8.32, PAL technologies, Glasgow, UK), which generated Event.csv files for each device. These files were processed in ProcessingPAL-V1.1 (Leicester, UK) using a validated algorithm to identify waking hours, extended non-wear periods (≥5 h) and invalid data [33]. Following processing, files were visually checked for plausibility of sleep/non-wear classification using heatmaps. If sleep and wake times looked unfeasible, the diaries were referred to for verification and when times differed by ≥2 h, the diary times were utilised [34]. Additionally, removals noted in the diary were inspected against heatmaps and the events window in the PAL analysis software (V8.10.8.32, PAL technologies, Glasgow, UK), and removed using the software if deemed plausible. Bouts were considered as “non-wear/sleep”, if ≥50% of it was within the period reported in the diary [35]. To minimise known errors with self-reported diary data, based on inspections of the data and previously used methods [36], we considered sitting/lying or standing bouts lasting ≥3 h without transitions as non-wear time.

Children wore the ActiGraph GT9X on their non-dominant wrist [37], to improve compliance [38]. Wrist-worn accelerometers have demonstrated good validity in comparison to hip-worn accelerometers [39]. The data was collected at a 30 Hz sampling rate [40] and summed over 5-sec epochs. ActiGraph (ActiLife V6.13.3) software was used to initialise, download and process files. Chandler wrist-based cut-points [41], applied to the vector-magnitude, were used to categorise MVPA (≥818 counts/5-secs) and TPA (≥162 counts/5-secs). Non-wear time, defined as ≥90 consecutive minutes of zero counts [42], was removed using the software.

Periods when children were at home were uploaded into both the ActiGraph and Processing PAL software and matched with time-stamped data, allowing home-based PA and postural behaviours to be generated, respectively. Days were considered valid, when the device was worn for ≥75% of the time at home [43]. In accordance with previous research [44], children with completed home diaries, and at least one valid day with ≥3 h of wear time at home were included in the analyses. Reported illness days were also excluded from the analyses. ActivPAL and ActiGraph data in minutes, were divided by wear time at home and multiplied by 60 to create the dependent variables conveyed as averages/h [45].

### 2.6. Children Personal Information and Anthropometric Measures

Anthropometric measurements were taken at the children’s respective schools. Stature and body mass were measured to the nearest 0.001 m and 0.1 kg, using a portable stadiometer (Seca 213, Hamburg, Germany) and electronic weighing scales (Seca 876, Hamburg, Germany), respectively, using standard anthropometric techniques [46]. Body mass index (BMI), and subsequently BMI z-scores, were derived using the WHO (World Health Organization) growth reference standard [47].

### 2.7. Objectively Measured House and Garden Size

Objective house and garden size for each postcode were measured using GIS techniques, AddressBase Premium (ABP) [48] and Ordnance Survey MasterMap (OSMM) [49]. For residences (min 4–max 82), we extracted building footprints from OSMM and filtered out non-residential buildings, defined by ABP. The process was repeated to determine garden size for residences (min 2–max 82), defined in OSMM Greenspace dataset [50]. To estimate house size, a median of the extracted building footprints was calculated and multiplied by the number of floors in each house. A median garden size was also calculated for each home in the postcode.

### 2.8. Additional Measures

Parents reported their age, gender, whether they own or rent their home, educational status (Some secondary school/Completed secondary school/Trade qualifications or apprenticeship/Diploma or certificate/University degree or higher), the pre-tax annual household income, postcode and the number of children at home. Season of measurement covered four categories: Winter (December–February), Spring (March–May), Summer (June–August) and Autumn (September–November). Due to missing data on income and educational status, Welsh Index of Multiple Deprivation (WIMD) scores, derived from postcodes, were used as an indicator of socioeconomic status (SES). The WIMD scores, consider eight domains of deprivation; employment; health; income; housing; community safety; access to services; education; the environment [51]. Small areas in Wales are ranked 1–1909, where 1 is the most deprived and 1909 is the least deprived. For descriptive purposes, tertiles of SES were generated based on WIMD scores; low (1–636), medium (636–1272) and high (1272–1909). Daylength for the participants’ respective cities during each monitoring day was obtained from a valid and reliable online resource [52]. Family preferences and priorities for activity within the home [30], as well as parental media rules [53] were collected via validated questions. Social and individual factors have been known to influence children’s sedentary and activity behaviours at home [23]; therefore, they could play an important role in associations with such behaviours and the home environment. To identify the confounding factors, the coefficients were computed from the statistical models prior to and following adjusting for each variable. Variables with the greatest influence on the coefficients on average were controlled for in the models [54]. These were parent-reported child and parent activity preferences at home, parent perceptions of the importance of active play at home for their child, and whether parents enforce a maximum h/day of screen-time rule.

### 2.9. Statistical Analysis

Consent and assent as well as activPAL, ActiGraph, physical and social environment data were received for 235 (100%), 207 (88%), 214 (91%), 213 (91%) and 207 (88%) children, respectively. Statistical analyses were conducted using SPSS (IBM SPSS Statistics Inc., Chicago, IL, USA; Version 25), where significance was set at ≤0.05. Influential outliers were replaced with the largest or second smallest value in observations [55] for overall media equipment (*n* = 1) and bedroom media equipment (*n* = 1) summary scores. The unadjusted associations between each of the physical environment variables and the five home-based outcomes (min/h spent sitting, standing, in TPA and MVPA and the number of sitting breaks/h) were examined using linear regression (Model 1). Model 2 adjusted for home ownership, raw WIMD scores, season of measurement, daylength and the number of siblings at home, as well as the BMI, age and sex of the child. Model 3 further adjusted for social environmental factors associated with children’s PA and sedentary time. A final model (Model 4) was run for each of the five outcomes, including all the significant variables (*p* ≤ 0.10) [56] from model 3 and adjustment variables to determine independent associations between physical environment factors and the child home-based outcomes. Paired t-tests revealed that the outcomes differed between weekday and weekend days. However, separate analyses had little effect on findings; thus, weekday and weekend days were combined.

## 3. Results

Descriptive statistics are provided in Table 1. The participating children had a mean age of 10.2 ± 0.7, and 55% were girls. Children spent 40.3 ± 5.9 min sitting (67%), 12.3 ± 4.2 min standing, 21.6 ± 4.7 min in TPA, 6.7 ± 2.3 min in MVPA, and had 7.0 ± 1.9 sitting breaks per hour at home. Most parents who completed the audit and questions were female (83%), owned their home (86%), held a university degree (54%) and lived in the highest SES location (59%). Homes (i.e., the overall plot, including house and outdoor space) were perceived to have medium houses (60%) which were not detached (64%) and large gardens (46%), they mostly had two floors (77%), and had on average four occupants, including two children. Most parents enforced a maximum h/day of screen-time rule (69%) and on average thought it was ‘important’ for their child to engage in active play at home, their child and themselves enjoyed sedentary and PA activities at home ‘about equal’ and ‘strongly agreed’ that their child had enough space to play inside the house and in the back garden. Homes had 11.5 ± 2.1 rooms/areas, 57% had an open plan living area and 52% of the children had a TV in their bedroom. Homes averaged 27.7 ± 18.3 PA equipment items, 19.6 ± 8.0 seated furniture items, 2.0 ± 2.1 musical instruments, 11.6 ± 4.7 media equipment items overall and 1.9 ± 1.7 in the primary child’s bedroom. Lastly, homes tended to have digital TV (82%), access to a movie/TV streaming service (77%) and 3–4 smartphones.

### 3.1. Associations between Physical Home Factors and Home-Based Sitting

When all the confounding factors were controlled for, home-based sitting was negatively associated with a detached house (−2 min/h, *p* = 0.03), an open plan living area (−2 min/h, *p* = 0.01), perceived house size (−2 min/h, *p* = 0.01) and musical instruments, and positively associated with the presence of a TV in the child’s bedroom (+2 min/h, *p* = 0.03), bedroom media and overall media equipment (Table 2, Model 3). Children spent one additional min/h sitting at home for every 13 media equipment points (*p* < 0.01) and seven bedroom media equipment points (*p* = 0.03), and one min/h less for every six musical instrument points (*p* < 0.01). In the final model, negative associations with house size (−2 min/h, *p* = 0.02), an open plan living area (−3 min/h, *p* < 0.01), musical instruments and the positive association with media equipment remained (Table 2, Model 4). Children spent one additional min/h sitting at home for every 13 media equipment points (*p* < 0.01) and one min/h less for every seven musical instrument points (*p* = 0.01).

### 3.2. Associations between Physical Home Factors and Home-Based Standing

After adjusting for all confounding factors, a detached house (+2 min/h, *p* < 0.01), perceived house size (+1 min/h, *p* = 0.02), an open plan living area (+2 min/h, *p* = 0.01) and musical instruments were positively associated, whereas media equipment was negatively associated with home-based standing (Table 3, Model 3). Children spent one additional min/h standing at home for every eight musical instrument points (*p* < 0.01) and one min/h less for every 17 media equipment points (*p* < 0.01). In the final model, a detached house (+2 min, *p* = 0.02), an open plan living area (+2 min, *p* = 0.01) and musical instruments remained positively associated, while media equipment remained negatively associated with home-based standing (Table 3, Model 4). Children spent one additional min/h standing at home for every 10 musical instrument points (*p* = 0.01) and one min/h less for every 17 media equipment points (*p* < 0.01).

### 3.3. Associations between Physical Home Factors and the Number of Home-Based Sitting Breaks

Following adjustment for all confounding factors, the number of home-based sitting breaks was negatively associated with digital TV (−1 transition/h, *p* < 0.01) and positively associated with objective garden size (*p* < 0.01) (Table 4, Model 3). The number of home-based sitting breaks was still negatively associated with digital TV (−1 transition/h, *p* = 0.01) and positively associated with objective garden size (*p* = 0.03) in the final model (Table 4, Model 4).

### 3.4. Associations Between Physical Home Factors and Home-Based TPA

When controlling for all the confounding factors, home-based TPA was negatively associated with media equipment and positively associated with an open plan living area (+1 min/h, *p* = 0.05) (Table 5, Model 3). Every 20 media equipment points (*p* = 0.01) was associated with one min/h less in home-based TPA. The number of floors in the house (+1 min/h, *p* = 0.04) and an open plan living area (+1 min/h, *p* = 0.04) were positively associated with home-based TPA in the final model (Table 5, Model 4).

### 3.5. Associations Between Physical Home Factors and Home-Based MVPA

Following controlling for all the confounding factors, home-based MVPA was negatively associated with media equipment, the number of smartphones at home and positively associated with an open plan living area (+1 min/h, *p* = 0.04) (Table 6, Model 3). Every 50 media equipment points (*p* = 0.03) and 1–2 increase in the number of smartphones at home (*p* = 0.01) were associated with one min/h less in home-based MVPA. In the final model, only the positive association between home-based MVPA and an open plan living area (+1 min/h, *p* = 0.05) remained (Table 6, Model 4). 

## 4. Discussion

This study identified several previously unexplored physical factors within the home as correlates of children’s sitting, standing and PA at home. An open plan living area, the number of floors, musical instrument accessibility and availability as well as objective garden size were significantly influential, although, given these relationships have not been investigated before, it is difficult to make comparisons with past work. This is one of the first in the field to use a posture monitor and to examine home-based PA and sedentary time and found that children spent 46% of their time at home, which reinforces the importance of investigating the correlates of PA and sedentary time in this environment.

The layout of the family home as open plan living, compared with a more segmented living space was shown in this study to be independently associated with less sitting, more standing, more TPA and more MVPA irrespective of demographic factors, the social environment and other significant home factors. According to qualitative research [23,57], the lack of dividing walls in open plan living areas enable parents to better monitor electronic media usage and enforce rules. Indeed, electronic media rules have been shown to be associated with lower screen-time in children [17,58]. Furthermore, open-plan design may also provide more space to accommodate alternatives to screen-based pursuits [57].

This study is the first to include a measure of the number of floors in houses, observing a significant positive association with TPA. Additional floors in houses may result in higher TPA via increased stair usage. Indeed, the energy cost of stair climbing in children is between 5.3 and 8.8 METs [26], which is considered moderate-to-vigorous intensity. However, the relationship did not reach significance until the final model, implying the relationship is mediated by other physical environmental factors associated with TPA. This would suggest that the number of floors in houses is not uniquely associated with TPA.

Our findings showed that increased perceived, but not objective, house size, was associated with less sitting. This may suggest that perceived and objective house size have differential effects on sitting, yet it may also be because of the way objective house size was measured. It is possible that the objective house size measure may not be a true measure of size, as it was not the exact house size, but instead the median size of houses in the same postcode unit. One previous study [59], reported no association between self-reported house size and sedentary time among Spanish children aged 9–18 years. This discrepancy may be due to the present study measuring home-based sedentary time, and not sedentary time across the entire day. Indeed, it might be that only home-based, not overall, sedentary time is influenced by house size. A study that examined the influence of spatial organisation in homes on activity found adults in houses with higher integration between rooms (greater interconnectedness) spent more time sedentary, particularly watching TV [60]. The mechanism proposed for this was that a greater interconnectedness between rooms encourages social interaction, which in turn can lead to increased time spent in sedentary activities that are susceptible to social life in homes such as TV viewing. Larger houses may have less interconnectedness overall, as they have more rooms, and the average connectivity between rooms does not increase in larger houses [60]. Although speculative, a higher interconnectedness amongst rooms in smaller houses may increase sitting time by prompting participation in social sedentary activities such as TV viewing.

Increased accessibility and availability of musical instruments was associated with less sitting and more standing at home, which is interesting as many musical instruments can be played sitting or standing [26]. Playing musical instruments may displace sitting activities, such as screen-time, studying, socialising, and increase standing periods. Future research should seek to investigate this relationship further, particularly given the cognitive benefits of playing a musical instrument [61].

There was a strong association between greater accessibility and availability of media equipment and reduced standing and increased sitting at home, which was robust to adjustment for social and demographic factors. In one of the few other studies to have a combined measure of the accessibility and availability of media equipment, a positive relationship was found with screen-time in girls, but not with overall sedentary time in either sex [19]. Most studies [19,62], but not all [59], have shown no association between household media equipment and overall sedentary time. Moreover, bedroom media equipment was positively associated with sitting, but not after adjusting for the other significant factors, in contrast to previous studies that have predominantly shown no association [63,64,65]. The present study used a posture monitor, whereas others have used accelerometery [16,64], which is considered a less accurate measure of sitting [66]. Whilst the lack of a relationship between bedroom media equipment and MVPA is congruent with previous research [63,64], some studies found contradictory results [67,68]. Such contradictory findings may be attributable to, at least in part, methodological differences and large inter-individual variation. Nonetheless, our findings highlight the important role the home media equipment environment may have by encouraging sitting and consequently reducing standing through acting as a prompt to engage in screen-time.

Despite the plethora of studies investigating the influence of media equipment, it is worth noting that, to our knowledge, only one previous study has measured home-based behaviour, whereby no relationship was found with bedroom media equipment and either sedentary time or PA in primary school aged children [16]. As behaviours are likely shaped by characteristics of the setting in which they occur, it is important to measure sedentary time and PA at home, to improve the understanding of the factors that influence these behaviours in this environment. Supporting this approach, screen-based behaviours, that most often occur at home [14], have been consistently positively associated with media equipment in the home [59,62] and in the bedroom [64,65]. Therefore, further research measuring home-based sitting and PA objectively may provide some clarity on the role of media equipment in influencing children’s PA and sitting.

Children with digital TV at home had fewer sitting breaks. Pay TV/digital TV has been associated with increased TV viewing in adolescents [69], and screen-time in pre-school children [70]. Therefore, a greater choice of TV channels may be compelling to children, keeping them entertained for longer periods, resulting in less frequent sitting breaks. In addition, objectively measured garden size was positively associated with sitting breaks. This would suggest that children with larger gardens have more opportunities for breaking up screen-based sedentary activities. Fittingly, objectively measured garden size was also positively associated with MVPA. However, the association was attenuated with the addition of the social factors to the model. This indicates that factors such as the importance parents place on their children engaging in active types of play and parental restrictions on screen-time explain why some children do more MVPA and have larger gardens.

Despite the inconsistencies in the literature, our findings demonstrate the potential efficacy of removing electronic media from bedrooms and limiting the electronic media presence in homes to reduce children’s sitting time. Given the association between greater accessibility and availability of musical instruments and reduced sitting and increased standing, encouraging children to learn a musical instrument requires exploration as a strategy for reducing children’s sitting. Considering the potential utility of an open plan living area in allowing parents to better monitor electronic media usage and accommodating alternatives to sedentary activities, moving electronic media to an area that permits parental supervision and reconfiguring furniture to create space hold promise as strategies for reducing children’s sitting time and increasing their PA. Our findings also suggest that larger gardens may be important for PA, and particularly for increasing sitting breaks. This is important, given there is emerging evidence that more frequent sitting breaks are beneficially associated with metabolic indicators in children [71], particularly when interrupted with moderate walking [72]. Therefore, strategies which break up prolonged sitting such as encouraging children to take 5-min walking breaks during adverts when watching TV or after completing a level while playing video games should be incorporated into an intervention. The provision of standing or PA breaks is a strategy that has been incorporated into school-based interventions, which successfully increased PA and decreased sitting [73].

More insight into the behavioural type and broader contextual information may lead to a better understanding of the determinants of PA and sedentary time at home. Automated wearable cameras when used alongside accelerometery and inclinometers could provide important information on where the behaviour occurs, as well as the type of behaviour being performed [74]. However, given participants may be wearing the device in situations unsuitable for photography, research involving this technology remains problematic [75]. Radiofrequency identification and open beacon proximity tags hold potential to assess the location of behaviours at home (e.g., bedroom, lounge or kitchen); however, such technology is currently expensive and difficult to implement in homes due to their weak Wi-Fi coverage [76], compared with environments where it has been trialled previously, such as offices [77] and cares homes [78]. Technologies that provide objective contextual information for sedentary time and PA at home will mostly likely be available for use in the imminent future.

This study has numerous strengths, such as the use of the comprehensive audit to measure the physical environment, the assessment of sitting and standing using a posture sensor, the home-based measures of behaviours and the exploration of several previously unstudied physical variables. Furthermore, a wide range of important mediating factors were controlled for and the high response rate increased the representativeness of the findings. We also included both perceived and objective measures of the environment, based on recommendations of several reviews [79], as they are related to behaviours differently [80]. Nonetheless, it is important to acknowledge the limitations. Some degree of misclassification of when the children were at home is likely, as we relied on self-reporting to determine this. However, there are currently no feasible objective alternatives for measuring children’s location-specific behaviours. Whilst the sample size was relatively small, it was large enough to provide reasonable statistical power [29]. Although this is one of the first studies to measure house and garden size objectively and investigate how they relate to children’s PA and sitting, since full home addresses were not available, we could only obtain measures for each postcode, and not for the specific homes. Thus, the measures may not reflect the true environments, as not all homes with the same postcode are identical. Additionally, total garden and house size may not correspond to usable space where children can be active and play. Whilst we tried to account for this by measuring actual space to play inside and outside via self-reporting, space syntax software could be used in combination with floor plans to measure indoor space [81] and also the degree of integration amongst rooms [60]. Furthermore, although beyond the scope of the current study, future work should also seek to explore these relationships during the school holidays, when children are less active and more sedentary [82]. Due to the cross-sectional nature of the study, causal relationships could not be inferred. Relationships may be complex, and it is likely that social factors work in combination with the physical environment to influence behaviours. Nonetheless, these findings are novel and add valuable knowledge to the evidence base.

## 5. Conclusions

In conclusion, the results suggest that some aspects of physical home environment may have an important influence on children’s sitting, standing and PA at home, even after adjusting for socio demographic and social environmental factors. Therefore, it is imperative that future interventions target this environment, especially given children in this study spent a large proportion of their time at home sitting (67%) and the lack of previous home-based interventions [17]. Based on the results, strategies such as reconfiguring furniture to increase space, introducing electronic media breaks, promoting time spent in the garden, and housing electronic media in areas which allow parental supervision could be effective. Given the known influence of the social environment [23], and the impact of the physical environment on sitting and PA, interventions that consider both factors hold most promise. Lastly, although several physical factors are not easily modified, the findings could help impact future home and planning design to reduce sitting and increase PA and to help promote healthy active living in families.

## Figures and Tables

**Table 1 ijerph-16-04178-t001:** Participant characteristics and descriptive statistics.

Variable	Mean (SD) or %	*n*
Parental Characteristics		
Parent age	41.5 (5.7)	211
Parent gender (% Female)	83%	213
Parent activity preferences at home ^2^	3.4 (0.7)	211
Parent perceived importance of engaging in active play at home for child^ 3^	4.0 (0.8)	207
Maximum h/day of screen-time rule (% yes)	69%	206
Parental education **		207
Secondary school or lower	12%	
Diploma/Trade	34%	
University degree or higher	54%	
Child Characteristics		
Child age	10.2 (0.7)	233
Child sex (% Girl)	55%	235
Child BMI-z-score	0.6 (1.1)	233
Child activity preferences at home ^2^	3.3 (0.8)	207
Family Characteristics		
Number of siblings	1.2 (0.9)	213
Number of people at home	4.1 (1.1)	213
Home ownership		213
Rent	14%	
Own	86%	
SES (based on WIMD scores) **		220
Low	14%	
Medium	27%	
High	59%	
Home Characteristics and Features		
Perceived house size		213
Small	8%	
Medium	60%	
Large	32%	
Objectively measured house size (m^2)^	145.0 (52.1)	207
Perceived garden size		213
No garden	1%	
Small	15%	
Medium	38%	
Large	46%	
Objectively measured garden size (m^2)^	269.0 (166.7)	214
Type of house		213
Detached	36%	
Not detached (semi-detached, terrace, bungalow, flat)	64%	
Number of floors		213
*1*	4%	
*2*	77%	
*>2*	19%	
Space to play		211
Inside the house ^1^	3.6 (0.7)	
Back garden ^1^	3.6 (0.7)	
Front garden ^1^	2.6 (1.2)	
Audit Variables		
Total number of Rooms/Areas **	11.5 (2.1)	210
Presence of a TV in the child’s bedroom (% yes)	52%	212
Number of living areas with a TV at home	1.5 (0.6)	210
Presence of an open plan living area (% yes)	57%	211
Equipment Variables:		
Number of PA equipment items**	27.7 (18.3)	210
PA equipment accessibility and availability score	86.7 (63.1)	209
Number of seated furniture items **	19.6 (8.0)	210
Seated furniture accessibility and availability score	76.5 (31.2)	209
Number of media equipment items **	11.6 (4.7)	210
Media equipment accessibility and availability score	44.2 (18.2)	209
Number of bedroom media equipment items **	1.9 (1.7)	212
Bedroom media equipment accessibility and availability score	6.9 (6.3)	210
Number of musical instrument items **	2.0 (2.1)	210
Musical instrument accessibility and availability score	7.2 (7.5)	209
Electronic Media Equipment		
TV service		213
Digital (e.g., SKY, BT etc.)	82%	
Freeview or other	18%	
Movie/TV streaming (e.g., Netflix, Amazon TV etc.) [% yes]	77%	213
Number of smartphones (mode)	3–4	213
Outcome Variables		
Home-based activPAL outcomes		207
Full days of activPAL wear at home	5.3 (1.1)	
h/full day of activPAL wear at home	5.8 (1.6)	
Min/h spent sitting, % of time at home*	40.3 (5.9), 67%	
Min/h spent standing, % of time at home*	12.3 (4.2), 21%	
Min/h spent stepping, % of time at home**	7.5 (2.8), 12%	
Number of sitting breaks/h	7.0 (1.9)	
Home-based ActiGraph outcomes		214
Full days of ActiGraph wear at home	5.5 (0.9)	
h/full day of ActiGraph wear at home	5.8 (1.6)	
Min/h spent in MVPA, % of time at home	6.7 (2.3), 11%	
Min/h spent in TPA, % of time at home	21.6 (4.7), 36%	

^1^ 1 = strongly disagree; 5 = strongly agree; ^2^ 1 = almost always—sedentary; 5 = almost always—PA; ^3^ 1 = unimportant; 5 = very important; * % = proportion of time at home; ** Displayed for descriptive purposes only.

**Table 2 ijerph-16-04178-t002:** Associations between physical home factors and children’s home-based sitting.

Variable	Model 1			Model 2			Model 3			Model 4		
	B (SE)	β	*p*	B (SE)	β	*p*	B (SE)	β	*P*	B (SE)	β	*p*
Perceived house size	−1.56 (0.71)	−0.16	0.03 *	−2.24 (0.75)	−0.23	0.01 *	−1.98 (0.77)	−0.20	0.01 *	−1.77 (0.77)	−0.18	0.02 *
Objective house size	−0.01 (0.01)	−0.05	0.52	−0.01 (0.01)	0.07	0.36	−0.01 (0.01)	−0.10	0.24	–	–	–
Detached house	−1.27 (0.89)	−0.10	0.15	−2.31 (0.92)	−0.19	0.01	−2.12 (0.94)	−0.17	0.03 *	−1.29 (0.93)	−0.10	0.17
Number of floors	−0.28 (0.95)	−0.02	0.77	−0.36 (0.96)	−0.03	0.71	−0.53 (0.97)	−0.04	0.59	–	–	–
Open plan living area	−2.39 (0.84)	−0.20	0.01 *	−2.58 (0.85)	−0.22	0.01 *	−2.43 (0.86)	−0.20	0.01 *	−2.62 (0.81)	−0.22	<0.01 *
TV in child’s bedroom	1.99 (0.84)	0.17	0.02 *	1.76 (0.88)	0.15	0.05 *	1.92 (0.89)	0.16	0.03 *	0.66 (1.15)	0.06	0.57
Number of living areas with TV	0.67 (0.68)	0.07	0.32	0.38 (0.69)	0.04	0.59	0.75 (0.70)	0.08	0.29	–	–	–
PA equipment ^1^	0.00 (0.01)	0.00	0.97	−0.01 (0.01)	−0.03	0.66	−0.00 (0.01)	−0.02	0.83	–	–	–
Seated furniture ^1^	0.00 (0.01)	0.01	0.94	−0.01 (0.02)	−0.03	0.74	−0.01 (0.02)	−0.03	0.70	–	–	–
Media equipment ^1^	0.08 (0.02)	0.26	<0.01 *	0.08 (0.02)	0.24	<0.01 *	0.08 (0.02)	0.25	<0.01 *	0.08 (0.03)	0.23	<0.01 *
Bedroom media equipment ^1^	0.18 (0.07)	0.19	0.01 *	0.14 (0.07)	0.15	0.05 *	0.15 (0.07)	0.16	0.03 *	0.01 (0.01)	0.01	0.90
Musical instruments ^1^	−0.11 (0.06)	−0.15	0.04 *	−0.14 (0.06)	−0.18	0.01 *	−0.18 (0.06)	−0.23	<0.01 *	−0.15 (0.06)	−0.19	0.01 *
Digital TV	0.94 (1.08)	0.06	0.39	0.82 (1.05)	0.06	0.44	1.06 (1.08)	0.07	0.33	–	–	–
Movie/TV streaming	1.26 (1.00)	0.09	0.21	1.14 (1.00)	0.08	0.26	0.90 (1.03)	0.06	0.38	–	–	–
Number of smartphones	0.62 (0.65)	0.07	0.34	0.89 (0.68)	0.10	0.19	1.14 (0.73)	0.12	0.12	–	–	–
Space to play inside	−0.87 (0.62)	−0.10	0.16	−0.59 (0.61)	−0.07	0.33	−0.57 (0.62)	−0.07	0.36	–	–	–
Perceived garden size	−0.32 (0.57)	−0.04	0.57	−0.17 (0.56)	−0.02	0.77	0.07 (0.57)	0.01	0.90	–	–	–
Objective garden size	−0.00 (0.00)	−0.10	0.17	−0.01 (0.00)	−0.14	0.08	−0.00 (0.00)	−0.12	0.13	–	–	–
Space to play in front garden	−0.12 (0.35)	−0.02	0.74	−0.01 (0.35)	−0.00	0.99	0.06 (0.35)	0.01	0.88	–	–	–
Space to play in back garden	−0.75 (0.58)	−0.09	0.20	−0.51 (0.57)	−0.07	0.38	−0.49 (0.59)	−0.06	0.41	–	–	–
										R^2^ (adjusted R^2^) 0.33 (0.25)		

* *p* ≤ 0.05 in model 1, 2 and 4; * *p* ≤ 0.10 in model 3. ^1^ Accessibility and availability equipment score. Model 1: Unadjusted models for each physical factor. Model 2: Model for each physical factor adjusting for child BMI, age and sex, and the number of siblings, home ownership, season, WIMD and daylength. Model 3: Model for each physical factor adjusting for child BMI, age and sex, and the number of siblings, home ownership, season, WIMD, daylength, child preferences for sedentary or PA activities, parent preferences for sedentary or PA activities, parent perception of the importance of their child engaging in active play and a maximum h/day of screen-time rule. Model 4: Final model including all significant physical factors from models 3, adjusting for child BMI, age and sex, and the number of siblings, home ownership, season, WIMD, daylength, child preferences for sedentary or PA activities, parent preferences for sedentary or PA activities, parent perception of the importance of their child engaging in active play and a maximum h/day of screen-time rule.

**Table 3 ijerph-16-04178-t003:** Associations between physical home factors and children’s home-based standing.

Variable	Model 1			Model 2			Model 3			Model 4		
	B (SE)	β	*p*	B (SE)	β	*p*	B (SE)	β	*p*	B (SE)	β	*p*
Perceived house Size	1.01 (0.51)	0.14	0.05 *	1.40 (0.53)	0.20	0.01 *	1.28 (0.55)	0.18	0.02 *	0.96 (0.54)	0.13	0.08
Objective house size	0.00 (0.01)	0.03	0.67	0.00 (0.01)	0.04	0.66	0.00 (0.01)	0.05	0.54	–	–	–
Detached house	1.36 (0.62)	0.16	0.03 *	1.97 (0.64)	0.22	<0.01 *	2.09 (0.66)	0.24	<0.01 *	1.61 (0.66)	0.18	0.02 *
Number of floors	−0.40 (0.67)	−0.04	0.55	−0.17 (0.68)	−0.02	0.81	−0.16 (0.69)	−0.02	0.82	–	–	–
Open plan living area	1.37 (0.60)	0.16	0.02 *	1.58 (0.61)	0.19	0.01 *	1.54 (0.62)	0.18	0.01 *	1.58 (0.58)	0.19	0.01 *
TV in child’s bedroom	−1.19 (0.59)	−0.14	0.05 *	−1.03 (0.62)	−0.12	0.10	−1.01 (0.64)	−0.12	0.12	–	–	–
Number of living areas with TV	−0.55 (0.48)	−0.08	0.25	−0.45 (0.49)	−0.07	0.36	−0.59 (0.50)	−0.09	0.24	–	–	–
PA equipment ^1^	−0.00 (0.01)	−0.01	0.88	0.00 (0.01)	0.03	0.69	−0.00 (0.01)	−0.03	0.68	–	–	–
Seated furniture ^1^	−0.00 (0.01)	−0.01	0.85	0.01 (0.01)	0.03	0.66	0.01 (0.01)	0.04	0.63	–	–	–
Media equipment ^1^	−0.06 (0.02)	−0.27	<0.01 *	−0.06 (0.02)	−0.24	<0.01 *	−0.06 (0.02)	−0.24	<0.01 *	−0.06 (0.02)	−0.24	<0.01 *
Bedroom media equipment ^1^	−0.11 (0.05)	−0.17	0.02 *	−0.07 (0.05)	−0.11	0.13	−0.07 (0.05)	−0.11	0.16	–	–	–
Musical instruments ^1^	0.09 (0.04)	0.17	0.02 *	0.10 (0.04)	0.19	0.01 *	0.12 (0.04)	0.22	<0.01 *	0.10 (0.04)	0.18	0.01 *
Digital TV	−0.97 (0.76)	−0.09	0.20	−0.78 (0.74)	−0.07	0.29	−0.88 (0.77)	−0.08	0.25	–	–	–
Movie/TV streaming	−1.18 (0.70)	−0.12	0.10	−1.20 (0.71)	−0.12	0.09	−1.04 (0.73)	−0.10	0.16	–	–	–
Number of smartphones	−0.64 (0.46)	−0.10	0.16	−0.67 (0.48)	−0.11	0.16	−0.61 (0.53)	−0.09	0.25	–	–	–
Space to play inside	0.37 (0.45)	0.06	0.41	0.07 (0.43)	0.01	0.87	0.08 (0.44)	0.01	0.87	–	–	–
Perceived garden size	−0.21 (0.40)	−0.04	0.60	−0.33 (0.40)	−0.06	0.41	−0.39 (0.41)	−0.07	0.34	–	–	–
Objective garden size	0.00 (0.00)	0.05	0.48	0.00 (0.00)	−0.06	0.41	0.00 (0.00)	0.07	0.40	–	–	–
Space to play in front garden	0.09 (0.25)	0.02	0.74	−0.04 (0.25)	−0.01	0.89	−0.06 (0.25)	−0.02	0.83	–	–	–
Space to play in back garden	0.31 (0.42)	0.05	0.46	0.10 (0.41)	0.02	0.80	0.11 (0.42)	0.02	0.80	–	–	–
										R^2^ (adjusted R^2^) 0.30 (0.23)		

* *p* ≤ 0.05 in model 1, 2 and 4; * *p* ≤ 0.10 in model 3; ^1^ Accessibility and availability equipment score. Model 1: Unadjusted models for each physical factor; Model 2: Model for each physical factor adjusting for child BMI, age and sex, and the number of siblings, home ownership, season, WIMD and daylength; Model 3: Model for each physical factor adjusting for child BMI, age and sex, and the number of siblings, home ownership, season, WIMD, daylength, child preferences for sedentary or PA activities, parent preferences for sedentary or PA activities, parent perception of the importance of their child engaging in active play and a maximum h/day of screen-time rule; Model 4: Final model including all significant physical factors from models 3, adjusting for child BMI, age and sex, and the number of siblings, home ownership, season, WIMD, daylength, child preferences for sedentary or PA activities, parent preferences for sedentary or PA activities, parent perception of the importance of their child engaging in active play and a maximum h/day of screen-time rule.

**Table 4 ijerph-16-04178-t004:** Associations between physical home factors and children’s home-based sitting breaks.

Variable	Model 1			Model 2			Model 3			Model 4		
	B (SE)	β	*p*	B (SE)	β	*p*	B (SE)	β	*p*	B (SE)	β	*p*
Perceived house Size	0.37 (0.23)	0.12	0.10	0.41 (0.23)	0.13	0.08	0.39 (0.23)	0.12	0.10 *	0.05 (0.25)	0.01	0.86
Objective house size	0.00 (0.00)	0.11	0.13	0.00 (0.00)	0.11	0.15	0.00 (0.00)	0.11	0.15	–	–	–
Detached house	0.13 (0.28)	0.03	0.64	0.11 (0.28)	0.03	0.70	0.11 (0.29	0.03	0.72	–	–	–
Number of floors	0.21 (0.30)	0.05	0.49	0.34 (0.29)	0.08	0.23	0.36 (0.29)	0.09	0.23	–	–	–
Open plan living area	0.08 (0.27)	0.20	0.78	0.03 (0.26)	0.01	0.91	−0.10 (0.27)	−0.03	0.71	–	–	–
TV in child’s bedroom	−0.73 (0.27)	−0.20	0.01 *	−0.43 (0.26)	−0.12	0.11	−0.37 (0.27)	−0.10	0.18	–	–	–
Number of living areas with TV	−0.25 (0.22)	−0.09	0.24	−0.20 (0.21)	−0.07	0.35	−0.20 (0.21)	−0.07	0.36	–	–	–
PA equipment ^1^	0.00 (0.00)	−0.01	0.86	0.00 (0.00)	0.00	1.0	−0.01 (0.00)	−0.02	0.81	–	–	–
Seated furniture ^1^	0.00 (0.01)	0.05	0.46	0.01 (0.01)	0.13	0.07	0.01 (0.00)	0.14	0.06 *	0.00 (0.01)	0.05	0.49
Media equipment ^1^	−0.02 (0.01)	−0.20	0.04 *	−0.01 (0.01)	−0.08	0.25	−0.01 (0.01)	−0.05	0.46	–	–	–
Bedroom media equipment ^1^	−0.04 (0.02)	−0.15	0.04 *	−0.02 (0.02)	−0.06	0.39	−0.01 (0.02)	−0.05	0.51	–	–	–
Musical instruments ^1^	0.02 (0.02)	0.09	0.22	0.02 (0.02)	0.07	0.33	0.01 (0.02)	0.05	0.49	–	–	–
Digital TV	−1.08 (0.33)	−0.23	<0.01 *	−1.11 (0.31)	−0.24	<0.01 *	−0.99 (0.32)	−0.21	<0.01 *	−0.86 (0.32)	−0.18	0.01 *
Movie/TV streaming	−0.33 (0.32)	−0.08	0.30	−0.10 (0.30)	−0.02	0.75	0.02 (0.31)	0.00	0.96	–	–	–
Number of smartphones	−0.33 (0.21)	−0.12	0.11	−0.29 (0.21)	−0.10	0.15	−0.24 (0.22)	−0.08	0.29	–	–	–
Space to play inside	0.49 (0.19)	0.18	0.01 *	0.36 (0.18)	0.14	0.05 *	0.35 (0.19)	0.13	0.06 *	0.35 (0.20)	0.13	0.08
Perceived garden size	0.16 (0.18)	0.06	0.39	0.19 (0.17)	0.08	0.26	0.16 (0.17)	0.07	0.35	–	–	–
Objective garden size	0.00 (0.00)	0.22	<0.01 *	0.00 (0.00)	0.23	<0.01 *	0.00 (0.00)	0.22	<0.01 *	0.00 (0.00)	0.16	0.03 *
Space to play in front garden	0.12 (0.11)	0.08	0.29	0.06 (0.11)	0.04	0.56	0.03 (0.11)	0.02	0.77	–	–	–
Space to play in back garden	0.30 (0.18)	0.12	0.10	0.29 (0.17)	0.12	0.09	0.26 (0.18)	0.11	0.14	–	–	–
										R^2^ (adjusted R^2^) 0.30 (0.22)		

* *p* ≤ 0.05 in model 1, 2 and 4; * *p* ≤ 0.10 in model 3; ^1^ Accessibility and availability equipment score; Model 1: Unadjusted models for each physical factor; Model 2: Model for each physical factor adjusting for child BMI, age and sex, and the number of siblings, home ownership, season, WIMD and daylength; Model 3: Model for each physical factor adjusting for child BMI, age and sex, and the number of siblings, home ownership, season, WIMD, daylength, child preferences for sedentary or PA activities, parent preferences for sedentary or PA activities, parent perception of the importance of their child engaging in active play and a maximum h/day of screen-time rule; Model 4: Final model including all significant physical factors from models 3, adjusting for child BMI, age and sex, and the number of siblings, home ownership, season, WIMD, daylength, child preferences for sedentary or PA activities, parent preferences for sedentary or PA activities, parent perception of the importance of their child engaging in active play and a maximum h/day of screen-time rule.

**Table 5 ijerph-16-04178-t005:** Associations between physical home factors and children’s home-based TPA.

Variable	Model 1			Model 2			Model 3			Model 4		
	B (SE)	β	*P*	B (SE)	β	*P*	B (SE)	β	*P*	B (SE)	β	*P*
Perceived house size	0.33 (0.57)	0.04	0.56	0.48 (0.58)	0.06	0.41	0.30 (0.59)	0.04	0.62	–	–	–
Objective house size	0.00 (0.01)	0.01	0.90	0.00 (0.01)	0.03	0.68	0.00 (0.01)	0.04	0.58	–	–	–
Detached house	−0.91 (0.70)	−0.09	0.20	−0.82 (0.70)	−0.08	0.24	−0.91 (0.71)	−0.09	0.21	–	–	–
Number of floors	1.04 (0.76)	0.10	0.17	1.20 (0.73)	0.11	0.10	1.28 (0.74)	0.12	0.09 *	1.48 (0.73)	0.14	0.04 *
Open plan living area	1.63 (0.67)	0.17	0.02 *	1.57 (0.65)	0.16	0.02 *	1.34 (0.67)	0.14	0.05 *	1.34 (0.66)	0.14	0.04 *
TV in child’s bedroom	−1.99 (0.66)	−0.22	<0.01 *	−1.10 (0.67)	−0.12	0.10	−1.04 (0.68)	−0.11	0.13	–	–	–
Number of living areas with TV	−0.83 (0.53)	−0.11	0.12	−0.80 (0.51)	−0.11	0.11	−0.93 (0.52)	−0.13	0.08 *	−0.79 (0.55)	−0.11	0.15
PA equipment ^1^	−0.01 (0.01)	−0.07	0.36	0.00 (0.01)	−0.00	0.97	−0.00 (0.01)	−0.02	0.78	–	–	–
Seated furniture ^1^	−0.01 (0.01)	0.03	0.63	−0.01 (0.01)	−0.03	0.64	−0.01 (0.01)	0.04	0.61	–	–	–
Media equipment ^1^	−0.07 (0.02)	−0.26	<0.01 *	−0.05 (0.02)	−0.19	0.01 *	−0.05 (0.02)	−0.18	0.01 *	−0.04 (0.02)	−0.13	0.10
Bedroom media equipment ^1^	−0.11 (0.05)	−0.14	0.05 *	−0.03 (0.05)	−0.05	0.53	−0.03 (0.05)	−0.04	0.56	–	–	–
Musical instruments ^1^	0.05 (0.05)	0.08	0.25	0.04 (0.04)	0.07	0.34	0.05 (0.05)	0.07	0.31	–	–	–
Digital TV	−1.06 (0.85)	−0.09	0.22	−1.06 (0.80)	−0.09	0.19	−0.93 (0.83)	−0.08	0.27	–	–	–
Movie/TV streaming	−1.35 (0.80)	−0.12	0.09	−0.78 (0.77)	−0.07	0.31	−0.53 (0.79)	−0.05	0.50	–	–	–
Number of smartphones	−1.21 (0.51)	−0.17	0.02 *	−1.04 (0.52)	−0.14	0.05	−0.96 (0.57)	−0.12	0.09	−0.45 (0.61)	−0.06	0.46
Space to play inside	0.59 (0.49)	0.09	0.23	0.25 (0.47)	0.04	0.59	0.13 (0.48)	0.02	0.79	–	–	–
Perceived garden size	−0.09 (0.45)	−0.02	0.84	−0.04 (0.43)	−0.01	0.93	−0.18 (0.44)	−0.03	0.69	–	–	–
Objective garden size	0.00 (0.00)	0.09	0.19	0.00 (0.00)	0.14	0.06	0.00 (0.00)	0.12	0.12	–	–	–
Space to play in front garden	0.00 (0.28)	0.00	1.00	−0.06 (0.27)	−0.02	0.82	−0.13 (0.27)	−0.03	0.63	–	–	–
Space to play in back garden	0.55 (0.46)	0.09	0.24	0.44 (0.44)	0.07	0.31	0.31 (0.45)	0.05	0.49	–	–	–
										R^2^ (adjusted R^2^) 0.28 (0.21)		

* *p* ≤ 0.05 in model 1, 2 and 4; * *p* ≤ 0.10 in model 3; ^1^ Accessibility and availability equipment score. Model 1: Unadjusted models for each physical factor. Model 2: Model for each physical factor adjusting for child BMI, age and sex, and the number of siblings, home ownership, season, WIMD and daylength. Model 3: Model for each physical factor adjusting for child BMI, age and sex, and the number of siblings, home ownership, season, WIMD, daylength, child preferences for sedentary or PA activities, parent preferences for sedentary or PA activities, parent perception of the importance of their child engaging in active play and a maximum h/day of screen-time rule; Model 4: Final model including all significant physical factors from models 3, adjusting for child BMI, age and sex, and the number of siblings, home ownership, season, WIMD, daylength, child preferences for sedentary or PA activities, parent preferences for sedentary or PA activities, parent perception of the importance of their child engaging in active play and a maximum h/days of screen-time rule.

**Table 6 ijerph-16-04178-t006:** Associations between physical home factors and children’s home–based MVPA.

Variable	Model 1			Model 2			Model 3			Model 4		
	B (SE)	β	*p*	B (SE)	β	*p*	B (SE)	β	*p*	B (SE)	β	*p*
Perceived house size	0.24 (0.28)	0.06	0.39	0.31 (0.28)	0.08	0.26	0.18 (0.28)	0.05	0.53	–	–	–
Objective house size	0.00 (0.00)	0.02	0.76	0.00 (0.00)	0.05	0.51	0.00 (0.00)	0.06	0.44	–	–	–
Detached house	−0.56 (0.34)	−0.12	0.10	−0.42 (0.33)	−0.09	0.21	−0.54 (0.34)	−0.11	0.11	–	–	–
Number of floors	0.52 (0.36)	0.10	0.15	0.50 (0.35)	0.10	0.15	0.57 (0.35)	0.11	0.11	–	–	–
Open plan living area	0.88 (0.32)	0.19	0.01 *	0.73 (0.31)	0.16	0.02 *	0.66 (0.32)	0.14	0.04 *	0.63 (0.32)	0.14	0.05 *
TV in child’s bedroom	−0.85 (0.32)	−0.19	0.01 *	−0.48 (0.32)	−0.11	0.13	−0.49 (0.32)	−0.11	0.13	–	–	–
Number of living areas with TV	−0.28 (0.25)	−0.08	0.27	−0.23 (0.24)	−0.06	0.36	−0.37 (0.25)	−0.11	0.13	–	–	–
PA equipment ^1^	−0.00 (0.00)	−0.03	0.67	0.00 (0.00)	0.03	0.66	0.00 (0.00)	0.01	0.91	–	–	–
Seated furniture ^1^	0.00 (0.01)	0.00	0.95	−0.00 (0.01)	−0.02	0.81	−0.00 (0.01)	−0.03	0.73	–	–	–
Media equipment ^1^	−0.03 (0.01)	−0.21	<0.01 *	−0.02 (0.01)	−0.15	0.03 *	−0.02 (0.01)	−0.15	0.03 *	−0.01 (0.01)	−0.08	0.28
Bedroom media equipment ^1^	−0.05 (0.03)	−0.14	0.06 *	−0.02 (0.03)	−0.06	0.36	−0.03 (0.03)	−0.07	0.31	–	–	–
Musical instruments ^1^	0.02 (0.02)	0.08	0.29	0.02 (0.02)	0.06	0.39	0.03 (0.02)	0.09	0.23	–	–	–
Digital TV	−0.43 (0.41)	−0.08	0.29	−0.37 (0.39)	−0.06	0.34	−0.40 (0.40)	−0.07	0.31	–	–	–
Movie/TV streaming	−0.48 (0.39)	−0.09	0.21	−0.36 (0.37)	−0.07	0.32	−0.27 (0.38)	−0.05	0.47	–	–	–
Number of smartphones	−0.63 (0.25)	−0.18	0.01 *	−0.60 (0.25)	−0.17	0.02 *	−0.69 (0.27)	−0.18	0.01 *	−0.49 (0.30)	–0.13	0.11
Perceived garden size	0.25 (0.22)	0.08	0.25	0.26 (0.21)	0.09	0.20	0.17 (0.21)	0.06	0.42	–	–	–
Objective garden size	0.00 (0.00)	0.10	0.16	0.00 (0.00)	0.16	0.03 *	0.00 (0.00)	0.13	0.07 *	0.00 (0.00)	0.09	0.20
Space to play in front garden	−0.12 (0.13)	−0.06	0.38	−0.14 (0.13)	−0.08	0.27	−0.17 (0.13)	−0.09	0.18	–	–	–
Space to play in back garden	0.25 (0.22)	0.08	0.26	0.21 (0.21)	0.07	0.33	0.15 (0.22)	0.05	0.47	–	–	–
										R^2^ (adjusted R^2^) 0.30 (0.23)		

* *p* ≤ 0.05 in model 1, 2 and 4; * *p* ≤ 0.10 in model 3; ^1^ Accessibility and availability equipment score. Model 1: Unadjusted models for each physical factor. Model 2: Model for each physical factor adjusting for child BMI, age and sex, and the number of siblings, home ownership, season, WIMD and daylength. Model 3: Model for each physical factor adjusting for child BMI, age and sex, and the number of siblings, home ownership, season, WIMD, daylength, child preferences for sedentary or PA activities, parent preferences for sedentary or PA activities, parent perception of the importance of their child engaging in active play and a maximum h/day of screen-time rule. Model 4: Final model including all significant physical factors from models 3, adjusting for child BMI, age and sex, and the number of siblings, home ownership, season, WIMD, daylength, child preferences for sedentary or PA activities, parent preferences for sedentary or PA activities, parent perception of the importance of their child engaging in active play and a maximum h/day of screen-time rule.

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
