# Peer review of "Associations between the Home Physical Environment and Children’s Home-Based Physical Activity and Sitting"

_ijerph, 2019, doi:10.3390/ijerph16214178_

Round 1
Reviewer 1 Report
This is a very interesting and extensive research project. It's a pity that the study didn't cover the whole year period. The omission of holiday months certainly influenced the overall results of the research - researchers should include this in chapter Limitation of this study.
The combination of both sets of data (results from working days and from weekends) in analyzes is surprising.
"Increased accessibility and availability of musical instruments was associated with less sitting and more standing at home, which is interesting as many musical instruments can be played sitting or standing [26]. Playing musical instruments appear to displace sitting activities, such as screen time, studying, socialising, and increase standing periods" - I have reservations about these conclusions, they are overinterpretation. "Although, the relationship between musical instruments and reduced sitting, and greater standing, requires exploration, it supports the possibility that encouraging children to learn how to play a musical instrument may be an effective strategy for reducing children’s sitting." -
encouraging to play a musical instrument is not a strategy to reduce the sediential lifestyle, eliminating the effects of physical inactivity; the authors draw too hasty conclusions (Discussion).
Author Response
"please see attachment"

Reviewer 2 Report
I enjoyed reading this comprehensive exploration of associations between the physical home environment and children's activity behaviors. I thought the authors did a really nice job drawing attention to relevant gaps in the literature and the statistical analysis was very thorough.
I have some very minor edits to suggest, which primarily serve to enhance clarity for the reader:
Line 227 reads "Homes were perceived to have medium houses (60%)" - should this read ''to be medium..."?
Line 234: ‘PA equipment items’ are referred to in the methods and results, but I can't find examples of these in either of those sections. Could you provide a few?
Lines 243-244: You have the following: “for every 13 media equipment (p = <0.01)” – is the ‘13’ a typo? Or is this supposed to say ‘for every 13 pieces of media equipment’?
In Table 1, in the ‘Audit variables’ section: I found it difficult to get a sense of the descriptives for some of the variables and I’m wondering if the authors can add some detail to the table to improve clarity and understanding for the reader. For example, PA equipment has a mean of 27.7… does this mean that people have an average of 27.7 pieces of PA equipment in their home? If so, perhaps the row title can be re-phrased to something like ‘Frequency of PA equipment’. Another example is the mean for ‘PA equipment and accessibility’ was 86.7… I went back to the methods section and re-read the information on this measure, but I’m still not sure whether if this is a high/low score… could the authors provide the range of possible scores to help with interpretation? If not, is there any other information that could be added to help with interpretation?
Line 335-336: To help readability, I think it would helpful to adjust the following text to this: "…less sitting, more standing, more TPA and more MVPA, irrespective of demographic factors…"
Line 354: This line reads ‘One previous study [59], reported no association among Spanish children aged 9-18 years.’ – I think it would be helpful to specify what they found no association between, even if it is implied in the previous sentences.
Author Response
"please see attachment"

Reviewer 3 Report
Had authors also obtained permission from each school and parental consents were obtained prior to conducting the study in each suburb although ethical approval obtained from university?
What were the inclusion and exclusion criteria of the study?
The HomeSPACE II audit had been validated in 2008. What were the audit assessed physical home environmental factors in the hypotheses as they influenced the children’s PA level and sedentary behaviors? The design and structure of the house and open living area?
How about the size of the home, the location of the school buildings and demographic data of the study sample as the participants were recruited from houses built in different suburbs? This concern was the same as the concern for the daylength in the participants’ city.
Same as for the recruited schools. What were the characteristics of the recruited schools?
How did it define the size of small, medium and large of house and back garden ? Why use postal code to measure the size of the house. There are small, medium and large houses in each suburbs.
How about the working parents who did not stay home and unable to complete the home log diary?
Was there any training or briefing session for parents and children to complete the home log diary? Were there any difficulties / barriers to fill in or complete home log diary by the children and parents?
Were all the parents capable to upload both the ActiGraph and ProcessingPAL sofeware.
Were all the children attending half day schooling or full day schooling at the time of data collection? This affected the children’s time at home.
Were there any study protocol for different schools to enhance the consistency in data collection and in analyzed the study data?
Statistical analysis:
Using three models for the adjustment of the physical home environmental factors, a final model (4) was run for each of the five outcomes including all the significant variables in model 3. Please explain why as there were discrepancies as shown in Table 2. What were the five outcomes?
What were the children’s home-based outcomes? Weekdays and weekends were different and why
Results: what was the mean and SD for parents enforced maximum screen-time rule for their children in this study? The demographics of the study participants should be elaborated as shown in Table 1.
What were the confounding factors for home-based sitting that had been controlled?
Study findings reported that larger garden size may associate with children’s sitting, standing and PA level, were there any play equipment placed in the garden?
Why the headings of 3.1 and 3.2 were the same as the findings showed that one was negatively associated and the other one was positively associated?
In the final model, the findings report that children spent one additional min/hr sitting at home for every 13 media equipment points and one min/hr less for every seven musical instruments points with significance level. Did one additional min per hour has any implications in study findings?
The authors did not provide details to report the findings for each table.
In the discussion, there were concerns with the increased perceived not objective house size that associated with less sitting and more standing. But the authors claimed that it was because the way house size was objectively measured, and it was not true measure. This could be avoided.
The study findings concluded that the physical home environment may have an important influence on children’s sitting , standing and PA level in the home design, but the present study reports that this is negatively associated with children’s standing, sitting and PA level.
Overall, there were too many factors to be considered without any clear operation definitions and study protocol to measure those physical environmental factors influencing children’s sitting, standing and PA level.
Author Response
|
Comments |
Authors’ response |
|
Had authors also obtained permission from each school and parental consents were obtained prior to conducting the study in each suburb although ethical approval obtained from university? |
Thank you for this important point. All schools provided Headteacher consent (Line 101; “Eleven schools (response rate 48%) consented”), alongside parental consent and child assent, (Line 105; “ Informed consent and child assent were provided”.) |
|
What were the inclusion and exclusion criteria of the study? |
Thank you for highlighting this omission. We have now included the study inclusion criteria on Lines 102-103, which reads: “To be eligible, children had to be aged 9-12 years and without a physical disability”. |
|
The HomeSPACE II audit had been validated in 2008. What were the audit assessed physical home environmental factors in the hypotheses as they influenced the children’s PA level and sedentary behaviors? The design and structure of the house and open living area? |
Thank you for raising this. The audit used was primarily based on the HomeSPACE-II validated in 2018, but included a measure of accessibility taken from the PA and media inventory from 2008. We have also validated the version we used, which is currently under review. The environmental factors are discussed on lines 112-132. |
|
How about the size of the home, the location of the school buildings and demographic data of the study sample as the participants were recruited from houses built in different suburbs? This concern was the same as the concern for the daylength in the participants’ city. |
Thank you. We measured house and garden size via self-report (line 119) and objectively (lines 186-191). We did not include the location of the schools, as we were only concerned with the home environment. Indeed, we only took the anthropometric measurements and provided children with the activity monitors in the schools. We have included demographic data in Table 1. |
|
Same as for the recruited schools. What were the characteristics of the recruited schools? |
Thank you. As per our response above, we were only concerned with the characteristics of the home environment and their influence on behaviour. As such, only anthropometric measurements physical activity monitoring data was conducted via the schools. |
|
How did it define the size of small, medium and large of house and back garden? Why use postal code to measure the size of the house. There are small, medium and large houses in each suburbs. |
Thank you. Parents were asked whether they think their house and garden is small, medium or large (line 119). They were also asked, on a 4-point likert scale (strongly disagree/disagree/agree/strongly degree) whether there was enough space for their child to play in the; (1) front garden; (2) back garden; and (3) inside the house (line 121). We have now provided descriptions of the independent variables as supplementary material. We used GIS to obtain objective measures of home size, as measuring 210 homes in person would be quite difficult logistically and ethically. |
|
How about the working parents who did not stay home and unable to complete the home log diary? |
Thank you. Most of the time, children had at least one parent with them after school to complete the diary. Where parents were unable to complete the log, the children completed it (see line 138). |
|
Was there any training or briefing session for parents and children to complete the home log diary? Were there any difficulties / barriers to fill in or complete home log diary by the children and parents? |
The participants were provided with instructions on how to complete the home log, which was very straightforward to complete. Participants only had to record the time the child left and returned home each day in a table. We have included the following sentence in the methods to clarify: “Instructions were provided, where “Home” was defined as a single location, including the house, garden, driveway and verge of the home where the child spends most of their time (i.e., excluding homes of other parents)” (see lines 135-137). |
|
Were all the parents capable to upload both the ActiGraph and ProcessingPAL sofeware. |
The objective devices (ActivPAL and ActiGraph) were retrieved from the participants at school, and the data was then processed by the lead researcher. |
|
Were all the children attending half day schooling or full day schooling at the time of data collection? This affected the children’s time at home. |
Thank you. All the children were of primary school age (aged 9-11 years), so they all attend full day schooling. |
|
Were there any study protocol for different schools to enhance the consistency in data collection and in analyzed the study data? |
Thank you for raising this important concern. For descriptive purposes, we have now split the sample into SES tertiles based on WIMD scores (lines 202-205 and table 1). Although we do have a high proportion of children in the high SES group, this, is comparatively less than most studies. |
|
Statistical analysis |
|
|
Using three models for the adjustment of the physical home environmental factors, a final model (4) was run for each of the five outcomes including all the significant variables in model 3. Please explain why as there were discrepancies as shown in Table 2. What were the five outcomes? |
Thank you. The five outcomes were home-based sitting, home-based standing, home-based sitting breaks, home-based TPA and home-based MVPA (see lines 223-224). We conducted four models for each outcome, with a table for each outcome (Tables 2, 3, 4, 5 and 6). |
|
What were the children’s home-based outcomes? Weekdays and weekends were different and why |
Thank you. The five child outcomes were home-based sitting, home-based standing, home-based sitting breaks, home-based TPA and home-based MVPA. We only examined weekday and weekend differences to determine whether separate analyses were required, however, given that the separate analyses did not significantly differ, data were pooled to strengthen the models. |
|
Results |
|
|
What was the mean and SD for parents enforced maximum screen-time rule for their children in this study? The demographics of the study participants should be elaborated as shown in Table 1. |
Thank you. Given that parents enforcing the rule provides binary data, we do not feel that providing a mean and SD for this variable would be very meaningful. We have therefore provided a percentage of parents who enforced the rule. |
|
What were the confounding factors for home-based sitting that had been controlled? |
Thank you. We controlled for the same confounding factors for each outcome. “Model 2 adjusted for home ownership, WIMD, season of measurement, daylength, the number of siblings at home, as well as the BMI, age and sex of the child. Model 3 further adjusted for social environmental factors associated with children’s physical activity and sedentary time. A final model (Model 4) was run for each of the five outcomes, including all the significant variables (p < 0.10) [56] from model 3 and adjustment variables to determine independent associations between physical environment factors and the children’s home-based outcomes” (lines 225-230). |
|
Study findings reported that larger garden size may associate with children’s sitting, standing and PA level, were there any play equipment placed in the garden? |
Thank you. We wanted to determine whether garden size, irrespective of what was in it, had any influence on children’s behaviours. We did create an outdoor play equipment variable, but it wasn’t associated with anything. As such, we only included PA equipment as a summary measure. |
|
Why the headings of 3.1 and 3.2 were the same as the findings showed that one was negatively associated and the other one was positively associated? |
Thank you. Section 3.1 includes associations with sitting, while section 3.2 includes associations with standing. We have ensured this is correct and clear in the manuscript. |
|
In the final model, the findings report that children spent one additional min/hr sitting at home for every 13 media equipment points and one min/hr less for every seven musical instruments points with significance level. Did one additional min per hour has any implications in study findings? |
Thank you. Yes, for every seven musical instrument points, participants spent one less min/hr sitting which was a significant finding, see line 261; “one min/hr less for every seven musical instrument points (p = 0.01)”. The result can also be found in Table 2, model 4. |
|
The authors did not provide details to report the findings for each table. |
Thank you. Sections 3.1, 3.2, 3.3, 3.4 and 3.5 include the written results for each table. |
|
Discussion |
|
|
There were concerns with the increased perceived not objective house size that associated with less sitting and more standing. But the authors claimed that it was because the way house size was objectively measured, and it was not true measure. This could be avoided. |
Thank you. Given that only the postcodes were available, rather than the specific addresses, we could only determine size for each postcode unit. We have acknowledged this as a limitation on lines 264-468; Although this is one of the first studies to measure house and garden size objectively and investigate how they relate to children’s PA and sitting, since full home addresses were not available, we could only obtain measures for each postcode, and not for the specific homes”.
|
|
The study findings concluded that the physical home environment may have an important influence on children’s sitting , standing and PA level in the home design, but the present study reports that this is negatively associated with children’s standing, sitting and PA level. |
Thank you. By the physical environment, we are referring to the various factors (Equipment, Size, space, open plan living area) within it, that were associated with the outcomes.We have now revised the sentence to ensure clarity on line 479: “the results suggest that some aspects of physical home environment may have an important influence on children’s sitting, standing and PA at home, even after adjusting for socio demographic and social environment factors. |
|
Overall, there were too many factors to be considered without any clear operation definitions and study protocol to measure those physical environmental factors influencing children’s sitting, standing and PA level. |
Thank you for raising this concern. We agree that more transparency with how the factors were derived is needed and have therefore incorporated more detail on lines 125-128; “In addition, summary scores that accounted for the accessibility and availability of PA equipment, seated furniture, overall media equipment, media equipment in the child’s bedroom and musical instruments were created by multiplying each item by their accessibility rating (A=1, B=2, C=3, D=4) and on lines 113-116; Briefly, the audit allowed the presence, amount and accessibility of 41 media(e.g., TV, computer, etc),musical (e.g., Drums, Piano, etc.), PA (e.g., Balls, Trampoline, etc.)and seated furniture items (e.g., sofa, desk etc.)” We have also provided detailed descriptions of the independent variables as supplementary material.
|
